# A novel antibody for the detection of alternatively spliced secreted KLOTHO isoform in human plasma

**Shreyas Jadhav**[1], **Sudipta Tripathi**[1], **Anil Chandrekar**[1], **Sushrut S. Waikar**[2], **Li-Li Hsiao**[1]*

**1** Renal Division, Department of Medicine, Brigham and Women's Hospital, Harvard Medical School, Boston, Massachusetts, United States of America, **2** Section of Nephrology, Boston University Medical Center, Boston, Massachusetts, United States of America

* lhsiao@bwh.harvard.edu

**Data Availability Statement:** All relevant data are within the manuscript and its Supporting information files.

## Abstract

αKlotho is primarily known to express as a transmembrane protein. Proteolytic cleavage results in shedding of the extracellular domain which enters systemic circulation. A truncated form of αKlotho resulting from alternative splicing of the *αKLOTHO* transcript exists and is believed to be secreted, thereby also entering systemic circulation. Existing ELISA methods fail to distinguish between the two circulating isoforms resulting in inconsistencies in assessing circulating αKlotho levels. We have exploited a unique 15aa peptide sequence present in the alternatively spliced secreted isoform to generate an antibody and show that it is able to specifically detect only the secreted Klotho isoform in human plasma. This finding will facilitate in distinguishing the levels of different circulating Klotho isoforms in health and disease and enhance their potential to serve as a biomarker for CKD and other conditions.

## Introduction

Since its serendipitous discovery as an aging related gene, *αKLOTHO* has gained significant attention owing to its potential role in aging, Chronic kidney disease (CKD) and cardiovascular diseases [1–6]. It was initially reported to only express in the distal tubular epithelial cells of the kidney [7, 8]. The complex aging-like phenotype observed in *Kl*[-/-] mice established its role as an aging regulator which in-part has led to continued interest in this gene [1, 9]. Indeed, several studies have reported the protective role of Klotho in reversal of disease mechanisms [10]. It has since been shown to be expressed in human artery and cardiac myocytes [6]. A subsequent study showed that Klotho is expressed ubiquitously including relatively lower expression observed in the parathyroid gland, placenta, prostate and small intestine [11]. The accumulating evidence suggests its role in playing a protective role in several organ systems.

Klotho is a single-pass transmembrane protein which acts as a co-receptor for FGF23 by forming a complex with FGFR1 [12]. This signalling in the distal tubular epithelial cells is what helps maintain normal phosphate levels [13, 14]. Klotho has also been reported to play a protective role in vascular calcification [5, 6]. Recent report has identified a role for Klotho in

**Funding:** The authors received no specific funding for this work.

**Competing interests:** The authors have declared that no competing interests exist.

cardiac fibrosis by modulating the TGFB and Wnt pathways [15]. It was also included as a factor in gene therapy that showed an improvement of aging-related disease outcomes, including chronic kidney disease [16].

The Klotho protein comprises two repeats of 440 aa in the extracellular region termed KL1 and KL2 domains. Proteolytic cleavage of the transmembrane Klotho results in shedding of the extracellular domain which is known to enter systemic circulation [17]. This circulating isoform called soluble Klotho (sKL) potentially acts as a humoral factor and likely functions independently of FGF23 [18]. This isoform has been successfully detected in blood and cerebrospinal fluid [19]. Multiple studies have reported the existence of another isoform of Klotho resulting from alternative splicing of the *αKLOTHO* transcript which is believed to be secreted thereby also entering systemic circulation [7, 20]. It comprises of the KL1 domain and a unique 15aa sequence at the c-terminus [21]. Based on the fact that this isoform also enters circulation, it is likely to impact signalling pathways independent of the transmembrane and sKL isoform. Also, there is a potential to detect this isoform in blood and other body fluids. Given its significance in modulating several disease mechanisms, the circulating isoforms present a unique potential to serve as markers of health and disease.

Currently available methods to detect the circulating isoforms are unable to differentiate between sKL and secKL as they utilize antibodies that would detect all isoforms containing the KL1 domain [22]. This has hindered the assessment of circulating Klotho levels, reliably. Our goal for this study is to develop a new methodology to detect secKL, which could add value to currently available tools in detecting circulating KL. It may also potentially serve as a biomarker for disease detection.

## Materials and methods

### ELISA procedures

For direct ELISA, 96-well microtiter plates (44-2404-21, Nunc) were coated with 1μg/ml either recombinant secreted KL (secKL), protein, aa 34–549 (ab84072, abcam) or recombinant soluble KL (sKL) protein, 34–981 (5334-KL-025, R&D systems) in 100 μl carbonate-bicarbonate buffer pH9.5 (C3041-50CAP, Sigma -Aldrich) and incubated overnight at 4˚C. Wells were then blocked in 100 μl blocking buffer (1% BSA in 1X PBS containing 0.1% v/v Tween20) for 1 hour at RT. HRP conjugated secKL Ab diluted in 100 μl Antibody dilution buffer (1% BSA in 1X PBS containing 0.05% v/v Tween20) was added to each well and incubated at RT for 1 Hour. Between each step plates were washed four times in washing buffer (1X PBS, 0.1% v/v Tween 20. The reaction was visualized by adding 100 μl chromogen TMB and substrate (555214, BD Biosciences) and incubated for a maximum of 30 minutes for color development and the reaction was stopped by adding 100 μl STOP solution (Invitrogen). Absorbance was measured at 450 nm using a plate reader.

For indirect ELISA similar protocol was followed except that after primary Ab (secKL) incubation, HRP labelled isotype specific IgG (A01856, GenScript) was added.

For sandwich ELISA, plates were coated with 2μg/ml secKL Ab in carbonate-bicarbonate buffer and incubated overnight at 4˚C. Wells were washed in washing buffer and blocked. Samples were diluted 1:4 in dilution buffer (1% BSA in 1X PBS containing 0.05% v/v Tween20) and added to corresponding wells (100 μl/well). The plate was then incubated at RT for one hour. Following washings, detection antibody from IDK kit (KR 1700, Immun Diagnostik, Germany) was added at the recommended dilution and incubated at RT for one hour. The plate was then processed as above.

### secKL Ab generation

Following peptide sequence was provided to GenScript (GenScript USA Inc. 860 Centennial Ave. Piscataway, NJ 08854) for antibody generation: SQLTKPISSLTKPYH

GenScript synthesized the peptide, immunized rabbits and provided purified polyclonal secKL antibodies.

### HRP labelling

HRP labelling was performed using the Lightning-HRP Conjugation Kit (701–0000, Novus Biologicals) following the manufacturers protocol.

### Reagents

Recombinant beta-Klotho protein (5889-KB-050, R&D systems), Recombinant Human TGF-beta 1 protein (240-B-002, R&D systems), Recombinant Human IL-6 protein (7270-IL-010/CF, R&D systems), Recombinant Human IL-6 protein (3718-FB-010, R&D systems).

### Statistics

Statistical analyses were performed using the GraphPad Prism version 8.0 software. Each result is a representative of three experimental replicates. Error bars are +/- standard deviation unless indicated otherwise. We applied 2-tailed Student′s *t* test, as specified in the legends. A p-value less than 0.05 was considered significant.

## Results

### Novel secKL Ab can detect recombinant secKL (rsecKL) protein

We first tested the ability of the novel secKL antibody to detect secKL protein in a direct ELISA setting. The different ELISA modalities are shown in S1 Fig. 96-well ELISA plates were coated with recombinant secKL (rsecKL) protein at 2µg/ml in carbonate buffer. HRP-labelled polyclonal secKL Ab was used as the detection antibody. As shown in Fig 1A, the antibody was able to detect the coated rsecKL protein over a broad dilution range ranging from 1:1,000 to 1:400,000. This clearly demonstrates that the novel secKL Ab is able to detect rsecKL protein. We further chose to test the antibody in an indirect ELISA setting (S1 Fig). The plates were similarly coated with rsecKL protein at 2µg/ml. secKL antibody was used at a concentration of 1µg/ml and an HRP-labelled isotype matched IgG was used to detect the signal. Again, as shown in Fig 1B, this combination was also able to detect the rsecKL protein over a broad dilution range of the secondary Ab (1:1,000 to 1:100,000). Together this clearly shows that the antibody can detect rsecKL protein in an ELISA assay.

### Novel secKL Ab shows specificity for secKL protein

Next, we tested whether this novel antibody can specifically detect secKL protein. We used our direct ELISA system to determine the specificity. Plates were coated with either rsecKL protein or recombinant soluble Klotho (rsKL) protein at 2µg/ml. Detection was carried out with the HRP-labelled secKL antibody. While the antibody was able to detect rsecKL protein over a broad dilution range (1:1,000 to 1:100,000) as shown in Fig 1C, the rsKL protein remained undetected at all dilutions tested. Similarly, we also used a lower dilution range (1:100 to 1:1000) to determine whether we could detect the rsKL. As Fig 1D shows, the secKL antibody successfully detected rsecKL protein but failed to detect rsKL. We next tested the most commonly used commercially available ELISA kit from IBL (#27998, Immuno-biological

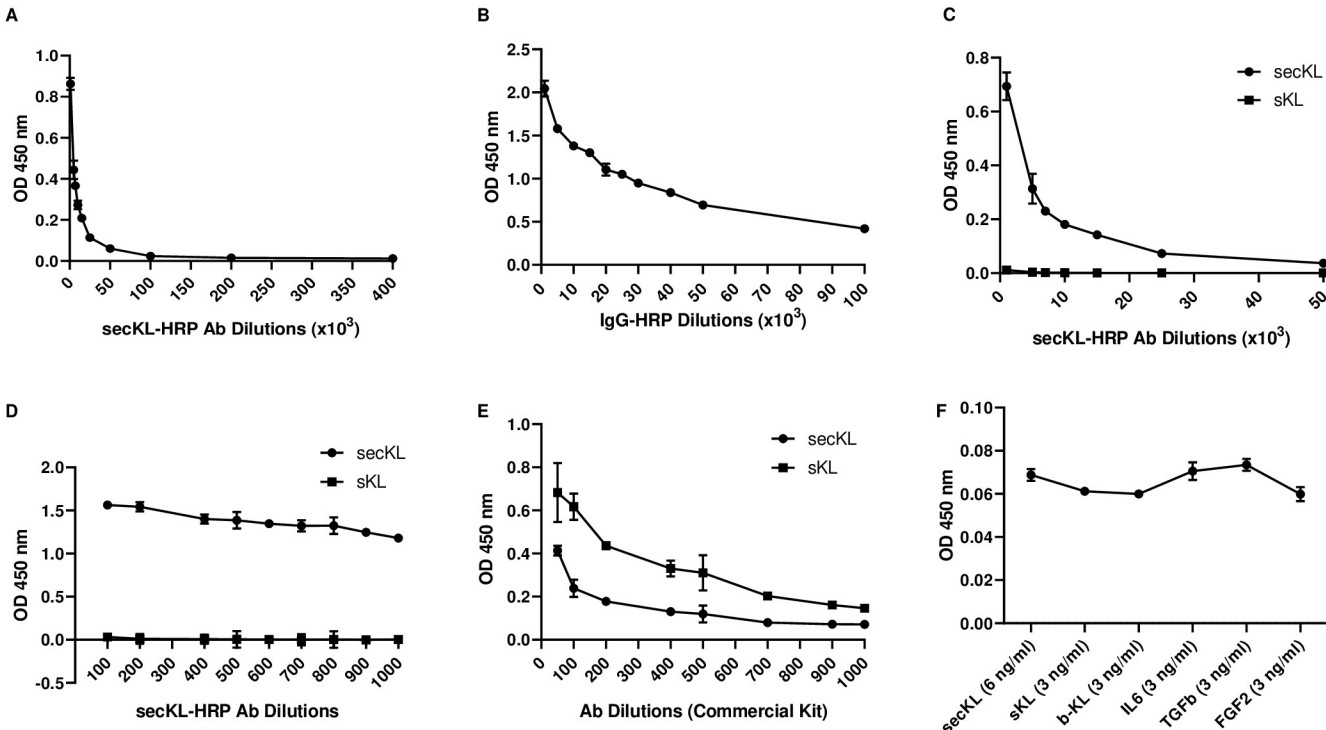

**Fig 1. High specificity of secKL antibody for secKL protein.** (A) secKL Ab detects rsecKL protein in a direct ELISA setting over a high dilution range:
1:1000, 1:5000, 1:7000, 1:10000, 1:25000, 1:50000, 1:100000, 1:200000, 1:400000 (B) secKL Ab detects rsecKL protein in an indirect ELISA setting via HRP
labelled secondary Ab (IgG dilutions: 1:1000, 1:5000, 1:10000, 1:15000, 1:20000, 1:25000, 1:30000, 1:40000, 1:50000, 1:100000) (C) secKL Ab can detect rsecKL
but not rsKL protein in a direct ELISA setting at a higher dilution range (Ab dilutions: 1:1000, 1:5000, 1:7000, 1:10000, 1:15000, 1:25000, 1:50000) (D) secKL
Ab can detect rsecKL but not rsKL protein in a direct ELISA setting at a lower dilution range (Ab dilutions: 1:100, 1:200, 1:400, 1:500, 1:600, 1:700, 1:800,
1:900, 1:1000) (E) Labelled detection antibody from IBL kit can detect both, rsKL and rsecKL proteins in a direct ELISA setting (Ab dilutions: 1:50, 1:100,
1:200, 1:400, 1:500, 1:700, 1:900, 1:1000) (F) secKL Ab can specifically detect rsecKL protein (6ng/ml) in the presence of indicated spiked-in non-specific
proteins (3ng/ml).

Laboratories, Japan), for its ability to detect the two circulating Klotho isoforms. ELISA plates
were coated with either rsecKL or rsKL protein and the HRP-labelled detection antibody from
the kit was used for detection. As seen in Fig 1E, the antibody from the kit was able to detect
both rsKL and rsecKL proteins at a dilution range from 1:100 to 1:1,000. We also determined
the specificity of the Klotho detection antibody from an additional commercially available
ELISA kit from IDK, following the manufacturers protocol. As seen in S2 Fig, this antibody
also detects both, rsecKL and rsKL proteins. We further tested the specificity of the secKL anti-
body in a sandwich ELISA setting (S1 Fig). Specifically, plates were coated with the novel
secKL antibody at 2µg/ml and rsecKL protein was added to the wells at 6ng/ml. Additionally,
wells were spiked with the indicated non-specific recombinant protein (Fig 1F) at 3ng/ml and
detection antibody from the IDK kit was used at the recommended dilution to detect the sig-
nal. As seen in Fig 1F, spiking-in with the indicated non-specific recombinant proteins, espe-
cially rsKL and recombinant beta KL, did not alter the detection of rsecKL by the novel Ab.
These results clearly demonstrate that the secKL antibody is able to specifically bind rsecKL
and not rsKL or rbetaKL. Also, antibodies from commercially available kits are unable to dif-
ferentiate between the various circulating Klotho isoforms. This likely explains the confound-
ing results seen for Klotho plasma levels in normal Vs disease states when detected using
commercial kits [23].

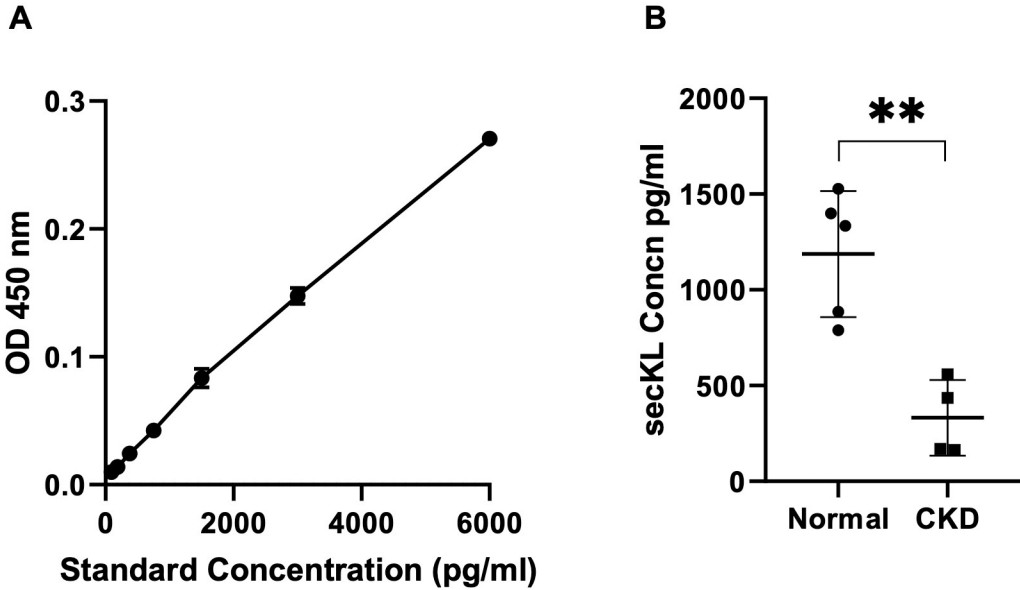

**Fig 2. secKL protein levels are reduced in human plasma from CKD samples.** (A) standard curve with secKL protein concentrations of 6000pg/ml, 3000pg/ml, 1500pg/ml, 750pg/ml, 375pg/ml, 187.5pg/ml, 93.75pg/ml (B) secKL protein levels in human plasma samples from normal (n = 5) Vs CKD (n = 4) (p< 0.003), confirmed that CKD is a state of Klotho deficiency. Two-tailed Student's *t* test applied (error bars, +/- standard error of mean).

## Novel secKL Ab can detect secKL protein in healthy and diseased human plasma samples

Given that the novel secKL antibody can selectively and specifically detect the rsecKL protein, we proceeded to test whether this Ab could also detect secKL protein in human plasma samples. We first determined the ability of the novel secKL antibody to detect secKL protein from either human plasma or urine samples in a sandwich ELISA setting using the secKL Ab as the capture Ab and labelled Ab from IDK kit as the detection Ab. The antibody detected secKL protein from normal human plasma but not urine (S3 Fig). We also performed a dilution series for human plasma and as shown in S4 Fig, the antibody is able to detect secKL protein over series of dilutions in a linear range. Finally, we proceeded to test human normal Vs CKD plasma samples in our ELISA setting with the novel secKL antibody as the capture antibody. Kidney function parameters for the samples are listed in S1 Table. The standard curve is as shown in Fig 2A. We successfully detected secKL protein in human plasma from normal and CKD samples. The levels of secKL protein were highly decreased in CKD plasma samples compared to normal (Fig 2B), the data is consistent with known reports showing that Klotho expression level is decreased in patients with CKD. This clearly indicates that the novel antibody can detect secKL protein from human plasma samples and is able to determine the differential levels of secKL protein from normal Vs CKD human plasma samples.

## Discussion

For decades there has been a lack of robust and credible detection methods for circulating Klotho. The existence of various isoforms and the extensive sequence similarity between them has further confounded the development of reliable detection methods. Given the significance of Klotho in health and disease and that it is present in systemic circulation, it is critical to develop detection methods that not only reliably detect but also distinguish between the known isoforms. We have successfully developed an antibody and a detection method that

specifically and selectively detects only the secreted Klotho isoform. Furthermore, the ability to detect secKL levels in human plasma samples and demonstrate the differential levels in normal Vs CKD samples shows immense potential for the novel antibody to present as a credible and reliable tool for the detection of circulating secKL protein. Generating a monoclonal antibody against the unique 15aa sequence will provide an antibody pair and further help in fine tuning the assay. Expanding the detection to a broader sample pool to further demonstrate the applicability of the antibody will reveal its true potential as a reliable biomarker.

To the best of our knowledge, this is the only study that shows the detection of secKL, specifically. There exists a mouse secKL antibody, K113, generated against the unique sequence but it has not been shown to detect secKL in blood or other body fluids [24]. Although mRNA levels for the full-length Vs secKl transcripts can be assessed to determine the relative expression levels of the two isoforms in normal Vs disease states, it does not always recapitulate protein expression levels. Also, given their distinct mechanisms of origin, there is a possibility that the sKL and secKL isoforms may not be present at the same level in circulation at any given point in time. As demonstrated in this study, the two most commonly used ELISA kits for Klotho detection do not distinguish between the two isoforms. They certainly appear to have a higher affinity for sKL in comparison to secKL. This might potentially skew the results thus obtained. In-line with this theory, a recent study aimed at determining the performance of assays to detect circulating Klotho reported some interesting findings. It showed that the immunoprecipitation-immunoblot (IP-IB) assay performed better at reliably detecting sKL levels over commercial ELISA assays [23]. Therefore, there is a need for better antibodies that will specifically detect the sKL isoform. As demonstrated in our study, currently available ELISA kits detect both sKL and secKL isoforms. This is likely because both these Klotho isoforms contain the KL1 domain and the antibodies from the commercial kits likely rely on the presence of the KL1 domain. The presence of multiple circulating isoforms and the inability of the antibody to specifically detect only one of these will continue to generate ambiguous results. Given that the secKL isoform is the only isoform with the unique 15aa sequence and the fact that it is potentially generated via a mechanism that is distinct from the other circulating isoforms, it might prove to be a more reliable readout in health Vs disease states. Also, the currently available methods are able to detect circulating Klotho from urine samples as well. Our Ab unambiguously detects secKL from plasma samples but did not detect any secKL protein from urine samples. We reason there could be at least two scenarios at play; 1) it might be that secKL protein is not shed in the urine of healthy subjects (the urine sample tested was from healthy subject) or 2) the amount of secKL shed in the urine is extremely low and therefore beyond the detection range of the Ab. Going forward, it will be ideal and more informative to develop the ability to assess the levels of these multiple circulating isoforms and determine whether there is any pattern in their expression in normal Vs disease states.

The lack of specificity for the various circulating isoforms by existing detection methodologies is perhaps the reason why they cannot reproducibly detect circulating Klotho. Our novel antibody nevertheless demonstrates the potential to develop a reliable detection method for the alternatively spliced secreted Klotho isoform. If we are able to detect the levels of these circulating isoforms distinctly, it might help to shed light on the yet unknown potential of this intriguing molecule. Perhaps secKL could prove to be a more reliable indicator of disease states and offer an opportunity for early intervention.

## Supporting information

**S1 Fig. A schematic depicting various types of ELISA modalities used.** In the direct and indirect ELISA assay, secKL protein was coated on the well and either labelled secKL Ab

(direct) or a labelled secondary Ab (indirect), in combination with the secKL Ab, was used to detect the signal. In the sandwich assay, secKL Ab was coated on the well and served as a capture Ab and labelled Ab from the IDK Kit was used as the detection Ab.
(TIF)

**S2 Fig. Labelled detection Ab from IDK kit detects rsKL, rsecKL and recombinant βKL proteins with different affinities.**
(TIF)

**S3 Fig. Novel secKL Ab detects secKL protein from human plasma and not urine samples.**
(A) standard curve with secKL protein concentrations of 6000pg/ml, 3000pg/ml, 1500pg/ml, 750pg/ml, 375pg/ml, 187.5pg/ml, 93.75pg/ml (B) seckL protein concentrations in human plasma and urine samples at indicated dilutions. While there is linear correlation in the plasma measurement, no protein detection observed in the urine, indicating that the Ab is able to detect differential expression in different specimens.
(TIF)

**S4 Fig. secKL protein detection by novel secKL Ab in normal human plasma samples at indicated serial plasma dilutions shows a lineal correlation.**
(TIF)

**S1 Table. Kidney function parameters for plasma samples.**
(DOCX)

## Author Contributions

**Conceptualization:** Shreyas Jadhav, Li-Li Hsiao.

**Data curation:** Shreyas Jadhav.

**Formal analysis:** Shreyas Jadhav.

**Funding acquisition:** Li-Li Hsiao.

**Investigation:** Shreyas Jadhav.

**Methodology:** Shreyas Jadhav.

**Project administration:** Shreyas Jadhav, Li-Li Hsiao.

**Resources:** Sudipta Tripathi, Anil Chandrekar, Sushrut S. Waikar, Li-Li Hsiao.

**Supervision:** Li-Li Hsiao.

**Validation:** Shreyas Jadhav.

**Visualization:** Shreyas Jadhav.

**Writing – original draft:** Shreyas Jadhav.

**Writing – review & editing:** Shreyas Jadhav, Li-Li Hsiao.

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
