## [Decision Letter · Decision Letter 0]

21 Oct 2020

PONE-D-20-19356

A novel antibody for the detection of alternatively spliced secreted Klotho isoform in human plasma

PLOS ONE

Dear Dr. Shreyas Jadhav,

Thank you for submitting your manuscript to PLOS ONE. After careful consideration, we feel that it has merit but does not fully meet PLOS ONE’s publication criteria as it currently stands. Therefore, we invite you to submit a revised version of the manuscript that addresses the points raised during the review process.

We look forward to receiving your revised manuscript.

Kind regards,

Ping-Hsun Wu, M.D.

Academic Editor

PLOS ONE

Journal Requirements:

2. In the Methods section, please provide the location of the GenScript company that synthesized the peptides for antibody generation.

Additional Editor Comments (if provided):

Re-plot the figures with consistent scales were advised, including Fig 1A, 1B, 1C, 1D, 1E, 1F, and 2A. The numbers of experimental replicates were recommended to demonstrate in the method section. Please revise as the reviewers' suggestions.

Reviewers' comments:

Reviewer's Responses to Questions

**Comments to the Author**

1. Is the manuscript technically sound, and do the data support the conclusions?

Reviewer #1: Yes

Reviewer #2: Yes

2. Has the statistical analysis been performed appropriately and rigorously? 

Reviewer #1: No

Reviewer #2: Yes

3. Have the authors made all data underlying the findings in their manuscript fully available?

Reviewer #1: Yes

Reviewer #2: Yes

4. Is the manuscript presented in an intelligible fashion and written in standard English?

Reviewer #1: Yes

Reviewer #2: Yes

5. Review Comments to the Author

Reviewer #1: The manuscript by Jadhav et al demonstrated the development and validation of a monoclonal antibody specifically recognize alternatively spliced secreted Klotho (secKL) in human plasma. The authors performed direct, indirect, and sandwich ELISA strategies to confirm the antibody recognized only the secKL but not the soluble Klotho (sKL), which is generated by proteolytic cleavage of transmembrane form of Klotho. The secKL contains a unique 15AA sequence at the C-terminal. Therefore, the authors showed that the successful generation of the 15AA-specific monoclonal antibody that could be used to specifically target secKL in human plasma samples.

Overall, the experimental design and results of this study is straightforward. The reviewer have some comments for the authors to consider:

1. The X-axis labeling in Fig 1A and 1B is likely mislabeled if the authors intend to show Log10 value of the Ab dilution factors. Also, the scale of the Y-axis for Fig 1A, 1B, 1C, 1D, 1E, and 1F are different. Some figures used Log10 OD450nm and some used OD450nm. It is suggested to re-plot the figures with consistent scales.

2. Similarly, the X-axis of Fig 2A is not correct.

3. In Fig 2B, the authors showed the serum secKL levels. What is the levels total KL (secKL+sKL) in these samples based on commercial KL kit? and what is percentage of secKL among total circulating KL?

4. It is not clear why the error bars are not shown in all Figures. How many experimental replicates were performed to generate the representative results? It is recommended to describe the information in figures legend or M&M section.

Reviewer #2: This is a clinically relevant paper that is likely to have translational relevance. Reliable antibodies that detect the secreted Klotho isoform in human plasma are required to assess it'a potential as a biomarker of age-related disease. Please rearrange the paper so that the figure legends go to the end rather in the middle of the results section. Statisatical analyses comparing the curves for figure 1C,D,E should be undertaken.

6. PLOS authors have the option to publish the peer review history of their article (what does this mean?). If published, this will include your full peer review and any attached files.

Reviewer #1: No

Reviewer #2: No

---

## [Author Response · Author response to Decision Letter 0]

1 Dec 2020

Dear Dr. Ping-Hsun Wu,

Thank you for your letter dated October 21, 2020 with comments. We greatly appreciate and agree with yours and the reviewers’ suggestions, and we have aimed to address all the comments in this letter and incorporated all suggestions and points into the paper. The changes have been highlighted in the revised manuscript for ease of reference. 

We hope that you will find favor in all the information provided. We would like to express our gratitude for your consideration of our manuscript and we look forward to hearing from you.

Thank you very much for your time and consideration.

Yours sincerely,

Li-Li Hsiao, MD, PhD, FACP

Director, Asian Renal Clinic

Director, Global Kidney Health Innovation Center

Co-Program Director, Harvard Summer Research Program in Kidney Medicine

Brigham and Women’s Hospital

Harvard Medical School

BLI Room 443, 221 Longwood Ave.

Boston, MA. 02115

617-525-7366 (o)

Phone: (617) 525-7366

Fax: (617) 525-7386

E-mail: lhsiao@bwh.harvard.edu

Point-by-point response to the editor’s and reviewers’ comments

Journal Requirements:

We thank the editor’s for pointing out our misstep. We have modified the manuscript following the recommended guidelines.

2. In the Methods section, please provide the location of the GenScript company that synthesized the peptides for antibody generation.

Per editor’s suggestion, we have updated this information in the “Materials and Methods” section Line 73-74, page 5

“Following peptide sequence was provided to GenScript (GenScript USA Inc. 860 Centennial Ave. Piscataway, NJ 08854) for antibody generation: SQLTKPISSLTKPYH”

Additional Editor Comments (if provided):

Re-plot the figures with consistent scales were advised, including Fig 1A, 1B, 1C, 1D, 1E, 1F, and 2A. The numbers of experimental replicates were recommended to demonstrate in the method section. Please revise as the reviewers' suggestions.

We thank you for your comments. We have re-plotted all graphs to maintain consistency in terms of the scales. In addition, each result shown in this manuscript is a representative of three replicates. This information has also been updated in the “Materials and Methods”, Line 85-87, page 5.

“Each result is a representative of three experimental replicates. Error bars are +/- standard deviation unless indicated otherwise.”

Reviewers' comments:

Reviewer's Responses to Questions

Comments to the Author

1. Is the manuscript technically sound, and do the data support the conclusions?

Reviewer #1: Yes

Reviewer #2: Yes

2. Has the statistical analysis been performed appropriately and rigorously? 

Reviewer #1: No

Reviewer #2: Yes

3. Have the authors made all data underlying the findings in their manuscript fully available?

Reviewer #1: Yes

Reviewer #2: Yes

4. Is the manuscript presented in an intelligible fashion and written in standard English?

Reviewer #1: Yes

Reviewer #2: Yes

5. Review Comments to the Author

Reviewer #1: The manuscript by Jadhav et al demonstrated the development and validation of a monoclonal antibody specifically recognize alternatively spliced secreted Klotho (secKL) in human plasma. The authors performed direct, indirect, and sandwich ELISA strategies to confirm the antibody recognized only the secKL but not the soluble Klotho (sKL), which is generated by proteolytic cleavage of transmembrane form of Klotho. The secKL contains a unique 15AA sequence at the C-terminal. Therefore, the authors showed that the successful generation of the 15AA-specific monoclonal antibody that could be used to specifically target secKL in human plasma samples.

Overall, the experimental design and results of this study is straightforward. The reviewer have some comments for the authors to consider:

1. The X-axis labeling in Fig 1A and 1B is likely mislabeled if the authors intend to show Log10 value of the Ab dilution factors. Also, the scale of the Y-axis for Fig 1A, 1B, 1C, 1D, 1E, and 1F are different. Some figures used Log10 OD450nm and some used OD450nm. It is suggested to re-plot the figures with consistent scales.

We thank the reviewer for their wonderful comments. We have re-plotted all the graphs to reflect consistency in the scales. 

The x-axis for Fig 1A and 1B has been updated to reflect the antibody concentrations. The scales for the Y-axis for Fig 1A,1B,1C, 1D, 1E and 1F have been re-plotted and are now consistent. All graphs now show OD450 nm on the Y-axes. 

2. Similarly, the X-axis of Fig 2A is not correct.

We thank you again for your suggestion. We have re-plotted the graph for Fig 2A with OD 450nm on the Y-axis and standard concentrations on the X-axis.

3. In Fig 2B, the authors showed the serum secKL levels. What is the levels total KL (secKL+sKL) in these samples based on commercial KL kit? and what is percentage of secKL among total circulating KL?

We thank the reviewer for this insightful comment. Our ultimate goal is to be able to detect both, secKL and sKL proteins and determine their differential levels in normal Vs disease states thereby attributing their contribution to either the normal or the disease state. In order to be able to reliably do this, we first need to determine the affinities for secKL Vs sKL proteins of the commercial antibody. At this point we do not have a clear understanding of whether the antibody has a preference for either isoform and whether the two isoforms compete with each other to bind to the antibody. Lacking this information, we cannot clearly determine, in the total circulating KL that the commercial antibody detects, what fraction each isoform represents. If the two isoforms were to be competing for the antibody (which is as yet unknown), it will further cloud the contribution from each isoform and therefore might mislead the final values, especially given the fact that their levels are altered in the disease state. 

4. It is not clear why the error bars are not shown in all Figures. How many experimental replicates were performed to generate the representative results? It is recommended to describe the information in figures legend or M&M section.

We thank the reviewer you for their comments. 

We have re-plotted the graphs to include error bars (+/- standard deviation). In addition, each result shown in this manuscript is a representative of three replicates. This information has also been updated in the “Materials and Methods” section, Line 85-87, page 5.

“Each result is a representative of three experimental replicates. Error bars are +/- standard deviation unless indicated otherwise.”

Reviewer #2: This is a clinically relevant paper that is likely to have translational relevance. Reliable antibodies that detect the secreted Klotho isoform in human plasma are required to assess it's potential as a biomarker of age-related disease. Please rearrange the paper so that the figure legends go to the end rather in the middle of the results section. Statistical analyses comparing the curves for figure 1C,D,E should be undertaken.

We thank the reviewer for their kind words on the “translational relevance” of this paper.

We apologize for the inconvenience. We have formatted the manuscript per the journal’s recommended guidelines, which suggests inserting the figure legend in the text right after the first mention of the figure. 

Fig 1C, 1D and 1E represent the specificity of the secKL antibody towards the secKL protein and the lack of specificity for the commercial antibody in terms of detecting both secKL and sKL proteins. While the secKL antibody is clearly able to detect the secKL protein over a wide range of dilution series, it is unable to detect the sKL protein over a similar dilution series under identical experimental conditions. This is clearly observed by the near zero OD values for the sKL protein, clearly indicating that the secKL antibody is unable to detect the sKL protein. On the contrary, the commercial antibody is able to detect both secKL and sKL proteins as seen by the non-zero OD values for both.

6. PLOS authors have the option to publish the peer review history of their article (what does this mean?). If published, this will include your full peer review and any attached files.

Do you want your identity to be public for this peer review? For information about this choice, including consent withdrawal, please see our Privacy Policy.

Reviewer #1: No

Reviewer #2: No

---

## [Decision Letter · Decision Letter 1]

5 Jan 2021

A novel antibody for the detection of alternatively spliced secreted Klotho isoform in human plasma

PONE-D-20-19356R1

Dear Professor Li-Li Hsiao,

We’re pleased to inform you that your manuscript has been judged scientifically suitable for publication and will be formally accepted for publication once it meets all outstanding technical requirements.

Kind regards,

Ping-Hsun Wu, M.D. PhD.

Academic Editor

PLOS ONE

Additional Editor Comments (optional):

All comments and suggestions were advised. The manuscript is accepted for publication.

Reviewers' comments:

Reviewer's Responses to Questions

**Comments to the Author**

1. If the authors have adequately addressed your comments raised in a previous round of review and you feel that this manuscript is now acceptable for publication, you may indicate that here to bypass the “Comments to the Author” section, enter your conflict of interest statement in the “Confidential to Editor” section, and submit your "Accept" recommendation.

Reviewer #1: All comments have been addressed

Reviewer #2: All comments have been addressed

2. Is the manuscript technically sound, and do the data support the conclusions?

Reviewer #1: Yes

Reviewer #2: Yes

3. Has the statistical analysis been performed appropriately and rigorously? 

Reviewer #1: Yes

Reviewer #2: Yes

4. Have the authors made all data underlying the findings in their manuscript fully available?

Reviewer #1: Yes

Reviewer #2: Yes

5. Is the manuscript presented in an intelligible fashion and written in standard English?

Reviewer #1: Yes

Reviewer #2: Yes

6. Review Comments to the Author

Reviewer #1: The authors have addressed my comments appropriately. One more suggestion is to move the supplementary figures to the main figures as this might increase readability.

Reviewer #2: The authors have addressed my concerns. The paper is correctly formatted. They have replotted the graphs including error bars. with corrected labelling of axis.

7. PLOS authors have the option to publish the peer review history of their article (what does this mean?). If published, this will include your full peer review and any attached files.

Reviewer #1: No

Reviewer #2: No

---

## [Editor Report · Acceptance letter]

11 Jan 2021

PONE-D-20-19356R1 

A novel antibody for the detection of alternatively spliced secreted KLOTHO isoform in human plasma 

Dear Dr. Hsiao:

I'm pleased to inform you that your manuscript has been deemed suitable for publication in PLOS ONE. Congratulations! Your manuscript is now with our production department. 

Kind regards, 

on behalf of

Dr. Ping-Hsun Wu 

Academic Editor

PLOS ONE